# Elasticities of Passenger Transport Demand on US Intercity Routes: Impact on Public Policies for Sustainability

Ignacio Escañuela Romana *, Mercedes Torres-Jiménez and Mariano Carbonero-Ruz

Departamento de Métodos Cuantitativos, Universidad Loyola Andalucía, Avda. de las Universidades s/n, Dos Hermanas, 41704 Sevilla, Spain; mtorres@uloyola.es (M.T.-J.); mcarbonero@uloyola.es (M.C.-R.)
* Correspondence: iescanuelaromana@al.uloyola.es

**Abstract:** Passenger transport is a key sector of the economy, and its sustainability depends on achieving the greatest possible efficiency, avoiding problems of congestion or underuse of infrastructures, and reducing the sector's environmental impact. Knowing the elasticities of demand is critical to achieving these objectives, estimating the intensity of transport demand, and predicting the effect of different policies on reducing greenhouse gas emissions. This research proposes a relatively simple model for estimating and predicting the elasticity of demand for different modes of transport at the route level. This model could be used by companies and public management to obtain a vision of the different analysed routes and the pressure of their demand, as well as a relative perspective of each of them. Such a model is used to estimate the price and income demand elasticities of passenger transport modes in domestic routes in the United States (2003–2019), where there is competition between road, rail, and air transport. Series of passenger numbers, fares, and budget shares are reconstructed from the available data. A Rotterdam demand model (RDM) is estimated using a seemingly unrelated regression method (SUR). The estimated income elasticities imply that demand for road transport increases somewhat more proportionally than the increase in income, somewhat less than proportionally for air transport, and with very low proportionality for rail transport. This indicates the need to target investment and service improvement efforts, as well as technological solutions, according to this difference in demand pressures. Finally, the demand response of the three modes of transport to price increases is inelastic, and there is little or no pass-through from one mode to another. This implies that fiscal or carbon pricing actions could have a very limited impact and high social costs. Again, strategies based on investments in technological progress, infrastructure development, and normative interventions could be more effective.

**Keywords:** passenger transport; modes of transportation; elasticities of demand; efficient and sustainable transport management

## 1. Introduction

The transport sector is very important in a country's economy. In the case of the US, it accounted for 9.1% and 8.7% of its GDP in 2018 and 2019 (11.6% in 1980), respectively (US Bureau of Transport Statistics, BTS [1,2]). In addition to this direct impact, transport modes and their costs are fundamental factors in understanding economic and human geography, as Krugman pointed out in his seminal 1991 article [3]. As stated by Rodrigue et al., transport is vital to the performance of an economy: "When transport systems are efficient, they provide economic and social opportunities and benefits that result in positive multiplier effects" [4].

On the other hand, there are major problems of passenger transport inefficiency and its impact in terms of noise emissions, pollutants, and greenhouse gases. Suffice it to give a brief overview of some of these problems. The US transport sector emitted 1836 million metric tons of carbon dioxide in 2022 (US Energy Information Administration, EIA [5]). The US total was 4964 million metric tons of carbon dioxide, thus making transport the

highest-emitting end-use sector. In this US transport service, light trucks (including SUVs, vans, and minivans) account for 37.1% of greenhouse gas emissions, passenger cars for 20.7%, commercial aircraft for 6.6%, other aircraft for 2.0%, ships and boats for 2.8%, and rail for 1.9% [6]. Global $CO_2$ emissions from the transport sector grew again in 2022 by 254 Mt [7].

Moreover, congestion and under-utilisation problems remain severe. Regarding congestion, different examples of problems can be cited: delays (increased fuel consumption, workers' overtime, passenger compensation); increased operational costs (congestion leads to inefficient routes and queuing patterns); reduced travel frequency or use of higher capacity means of transport, impacting on overall operational efficiency; infrastructure costs (higher investment in infrastructure to avoid congestions); and environmental impact (congestion can lead to increased fuel consumption and greenhouse gas emissions). A variety of different costs stem from overcapacity in passenger transport, including opportunity costs (revenue lost due to oversupply and unrealised demand) and inefficiency (variable cost per passenger increases with a lower margin if prices do not change, and there can be lower profitability due to the need to lower prices to increase demand).

The costs associated with some of those problems have been analysed in the literature. Thus, for example, Schrank et al. [8] estimated that congestion in road transport generated losses of USD 190 billion (2020 USD) in 2019 due to additional fuel costs and extra travel hours. This is consumption that increases greenhouse gas emissions: "the stop-and-go nature of congestion increasing emissions yet further" (Grote et al. [9], p. 95). Congestion is thus an important factor in road traffic emissions (op. cit.). The same problem occurs in air traffic, since, as the following was pointed out by Clarke et al. [10]: "Because of congestion, aircraft are often forced to fly far from the cruise altitude and/or the cruise speed for which they are designed. Such sub-optimality results in unnecessary fuel burn and gaseous emission" (p. 4).

Furthermore, the US Travel Association [11] deduced that traffic congestion caused Americans to avoid 47.5 million road trips in 2018, which cost the economy nearly USD 30 billion in lost travel spending. Moreover, the cost of delays in 2019 was estimated at USD 33 billion by a Federal Aviation Administration (FAA)-sponsored study [12]. Additionally, the Travel Industry Association's estimate of unrealised air travel in 2007 was valued at more than 41 million trips, which would have resulted in additional revenue of more than USD 26 billion (Travel Impact Newswire [13]). In turn, the problem of airport congestion was quantified in 2008 by Gelhausen et al. [14], who estimated that 6% of all flights are operated from capacity-constrained airports in the United States. Moreover, as some authors have argued (Burghouwt et al. [15]), congestion at airports leads to higher fares for air passengers. Finally, among many other issues raised, there is a current trend for airlines to use larger aircraft to cope with transport from congested airports (Pollard [16]). At the same time, the use of regional airports, currently less in demand, is likely to increase, based on smaller, less noisy, and more environmentally sustainable aircraft (Banchik et al. [17]). The development of airports and the aircraft type composition of airlines will depend on the evolution of demand and therefore on making a correct demand forecast (NASA Team [18]). In general, as Severino et al. [19] state, "transport systems efficiency plays a key role for communities' liveability and economy, being, in addition, an important factor in the economic integration of countries" (p. 1).

Ultimately, these problems highlight the need to make estimates and projections of the intensity of each of the modes of passenger transport and, consequently, of demand (Nar y Arslankaya [20]). There are several alternative methods of transport demand forecasting. Very important aspects of current study methods include the use of a large quantity of data collected through smart systems (e.g., use of smart cards, use of payment cards, use of numerical systems to quantify public transport) and, based on artificial intelligence, efforts to make predictions of future demand behaviour. This introduces greater variation of estimation and prediction methods compared to the situation prior to the development of digital traffic management systems and data processing. For example,

Nar and Arslankaya [20] use machine learning algorithms. Çelebi et al. [21] use neural networks to develop short-term passenger demand forecasting models. Qin et al. [22] note that recurrent neural networks, convolutional neural networks, and other methodologies are used in road traffic demand management. Orlando et al. [23] propose spatial models (a modern variation of the old gravity models) and the use of digital public transport data to predict the future frequency of public transport trips. However, all these methodological proposals face the problem of the difficulty of obtaining reliable passenger transport data. One of the most relevant ways of quantifying demand in the passenger transport sector is based on the estimation of its elasticity with respect to income and prices of the most relevant modes. Therefore, quantifying the elasticity of demand for the routes where transport is concentrated is vital to understanding the impact of price changes and income on all the issues outlined above. Li et al. [24] pointed out that "studying the price elasticity of demand is the only way to formulate scientific road pricing", which can be extended to all other modes. Moreover, the great relevance of estimating the elasticities of demand is reaffirmed in the scientific literature. For example, Zeng et al. [25] point out that the research into the price elasticity of demand for travel modes "has become a research hotspot." (p. 3). In addition, "the sensitivity of demand concerning the monetary price of travelling determines the supplier's ability to raise revenues by setting fares above the marginal social cost" (Hörcher and Tirachini [26], p. 2). Above all, forecasting through knowledge of elasticities would provide insight into reactions to future price and income developments, adding an understanding of potential consumer behaviour in different scenarios.

Finally, the movement of passenger demand between modes of transport is an important element in an emission reduction strategy, as rail transport implies lower emissions (e.g., Gama [27], Zeng et al. [25]). However, whether such a shift in demand between modes occurs depends, in part, on the value of the cross-price elasticities (the higher their positive value is, the stronger the response to price changes will be).

Although the estimation of passenger transport demand elasticities has been widely addressed in the scientific literature (Oum et al. [28], Goodwin [29], Holmgren [30], Gundelfinger [31], etc.), the results reveal wide ranges of these estimates (see Appendix C), meaning that further research is needed to illuminate this issue and approximate this value more accurately. Moreover, there is little recent evidence on the price elasticity of demand for public transport services (Davis [32]).

Consequently, this study focuses on demand elasticities, an important contemporary scientific research object (Zeng et al. [25]), and does so with respect to some of the most important routes in the United States. One of the advantages of estimation at the route scale is that it allows for differentiation between distinct cases. For example, factors such as distance, infrastructure, demographic and economic structure, etc., vary from one route to another, and an estimation at the national market level would exclude important existing differences.

Therefore, the present study has a general objective, which is to propose a relatively simple model that can be used to estimate and predict passenger transport demand on different routes, factoring in the various transport modes and making assessments based on accessible data from public statistical agencies. In this way it could be used, without complexity, by companies and public management to obtain a global view (through average values) of the different routes and the pressure of their demand, as well as a relative perspective of each of them, allowing for a comparison between routes. To achieve this objective and to show that it is feasible, the present study includes three intermediate aims, namely, (i) to contribute to estimating the elasticities of demand in passenger transport for domestic routes in the United States, where there is competition between air, road, and rail transport; (ii) to contribute to the comparison of those estimates on the different routes analysed from 2003 to 2019 in order to check whether there are relevant variations between them (which requires comparison on a common basis); and (iii) to draw some conclusions based on the relationship between the estimated demand elasticities and the effectiveness

of different public policies to be implemented to improve the sustainability of passenger transport in the United States on domestic routes.

In order to achieve these objectives, 20 routes in the United States have been selected based on the relevance of their passenger volume. Demand elasticities have been estimated using a procedure similar to that applied by Escañuela [33] when measuring the demand elasticity of the Northeast Corridor of the United States. This study expands the geographic area of study and thus provides a more general and homogeneous view of passenger transport demand in the United States. To obtain this estimation, road passenger data series, for which no accurate records exist in the United States, have been approximated. Making this information available can be seen as one of the indirect contributions of this study.

Demand is measured at the route level as the number of passengers using each mode of transport, meaning that the estimate is conditional on the level of expenditure within the group. A theoretical demand model, the Rotterdam demand model (RDM), which was initially proposed by Barten [34] and Theil [35], is applied. The RDM fulfils the different maximisation conditions (whose corresponding theoretical restrictions are detailed in point 2.2 of the methodology section), contributing to the results' robustness while enabling a comparison between routes.

## 2. Materials and Methods

The methodological body of this research is composed of four main elements: (i) an RDM imposed on a conditional level estimation; (ii) an assessment based on the assumption of a rational demander, which is later used to test the hypotheses derived from it; (iii) an estimation carried out at a conditional level; and (iv) a seemingly unrelated regression (SUR) model applied as a multi-equation estimation procedure.

### 2.1. Methods

#### 2.1.1. The Choice of the RDM for Estimation

Two main options have been found in the scientific literature regarding the empirical estimation of passenger transport demand. Specifically, there is one that uses aggregate data and values across the population and one that uses disaggregated values at the individual or household level (Dunkerley et al. [36], Oum [37]). Aggregate models rely on data to describe the behaviour of large groups of travellers (Winston and Small [38]). Different disaggregate models can be used to understand the behaviour of the transport service's user. Winston and Small [38] emphasize some of the advantages of disaggregated models, such as a greater number of observations, a greater number of attributes considered in the demanded service, a theoretical basis, and an explicitness of the sources of random disturbances. The key contributor to these models, on which much of the scientific literature on this topic is based, is McFadden [39], who started from an individual utility function on alternatives, each of which is based on a series of attributes. Among these models, the use of a discrete choice logit model could be an important alternative to the approach adopted in this paper. For example, with an object of study similar to that of the present work, Cirillo and Hetrakul [40] studied an optimal pricing strategy for the Acela Express (Amtrak) service using a multinomial logit model. Therefore, these methods involve obtaining primary data through the design of homogeneous questionnaires, allowing the subsequent comparison of results on different routes and at different points in time (Fillone et al. [41], Al-Salih and Esztergár-Kiss [42]). The current study analyses 20 national transport routes for which there is no standard source of information. Explaining and predicting aggregate phenomena on 20 different routes makes the aggregate level preferable. However, both because of conformity in the level of data between the model and the cases studied, and because of the difficulty of obtaining sufficient and homogeneous disaggregated data available for all the cases studied, the choice between aggregate and disaggregate models depends largely on the purpose of the study (Oum [37]). In our methodological approach,

secondary data are used, which are easy to obtain and are continuously available over time (annually).

Another important choice to be made concerns the application of a parametric or nonparametric method to estimate the elasticity of demand. Several examples of the application of nonparametric methods can be found in the literature. Thus, for example, Lago et al. [43] conducted a comparative study to estimate demand elasticities for transit services; Davis [32] applied the same methodology to periods of fare changes in Mexico City, Guadalajara, and Monterrey to estimate the price elasticity of demand for urban rail transport; and Hoang-Tung et al. [44] compared ex-ante and ex-post scenarios related to the introduction of a bus rapid transit service in Hanoi. The quasi-experimental method does not allow for the estimation of revenue or own-price and cross-price elasticities, whether uncompensated (Marshall) or compensated (Hicks), in the 20 studied routes. The non-parametric, quasi-experimental, or natural experimental methods require a change or shift in supply, on a stable demand curve, to estimate the value of one of the elasticities of one of the alternative transport modes. Therefore, we have chosen the parametric method since it allows us to capture all the elasticities.

Finally, different aggregated demand models, both theoretical and empirical, have been applied in the scientific literature (Deaton [45], Barnett and Seck [46]). However, current economic theory has not been able to answer the question regarding which specification is best for estimating the demand function in a particular dataset (Barnett and Seck [46]). In the present study, the selection of the RDM is based on two important advantages. The first is related to its theoretical basis. The problem of aggregation in the demand function is to find a valid way to relate a function that aggregates consumers and relates to the individual demand functions of each consumer. Therefore, the problem of consumer aggregation concerns the characteristics or properties the function retains when aggregating multiple individual consumers. The aggregate demand function only retains the properties of continuity and homogeneity and does not take any other properties from the individual functions (Varian [47]). However, while it is true that the aggregation problem also arises in the RDM as it does in other systems of demand equations (McFadden [48], Yoshihara [49], Barten [50]), this problem can be overcome. According to Barnett [51], the problem is solved when starting from weak assumptions with unproblematic validity by using the probability bounds of the Slutsky equations as the number of consumers increases. Compared to its main theoretical alternative, the Almost Ideal Demand System (Deaton and Muellbauer [52]), the RDM appears better at recovering the true values of the elasticities. As outlined by Barnett and Seck [46], this is true "when we implement exact aggregation within weakly separable branches of a utility tree." (p. 821). The second advantage is econometric, arising from the fact that the RDM determines its variables in terms of first differences, which is desirable in the presence of autocorrelation. As Granger and Newbold [53] recommended, taking the first differences of all variables should considerably improve the interpretability of the coefficients and avoid spurious regression. Moreover, although it is an old technique, RDM is still a useful and commonly applied model today (Nguyen et al. [54], Muhammad and Countryman [55], Clements et al. [56]).

### 2.1.2. The RDM and the Theoretical Conditions of Rational Consumer Theory

The RDM can be considered to be consistent with a variety of consumers' preferences since the utility function is not specified explicitly (Clements and Gao [57]). The mathematical core of the RDM (following Barnett and Serletis [58], p. 69) is given by a differential demand system with relations given in absolute (undeflated) prices, as shown in (1):

$$\overline{\omega}_{it} Dx_{it} = k_i + \theta_i DQ_t + \sum\nolimits_j \pi_{ij} Dp_{jt} + u_{it} \tag{1}$$

where ij are goods or services, i, j = 1, ..., n (in this case, they represent the different pairs of transport modes being compared, where n = 2 or 3 modes); t is the period; x is the quantities; p is the prices; k is a constant coefficient capturing the possible variation in tastes and other factors over time (Clements and Johnson [59]); and $Dx_{it}$ means $\ln(x_i) - \ln(x_{i(t-1)})$.

$DQ_t$ is the change in real income (as measured by applying a Divisia volume index that uses budget shares as weights):

$$DQ_t = \sum_{j=1}^{n} \overline{\omega}_{jt} Dx_{jt} \tag{2}$$

where $\omega_i$ i is the budget share of i. $\theta_i$ is the marginal budget share of the i use of money income (Barnett and Serletis [58]):

$$\theta_i = \omega_i \eta_{iy} = \frac{p_i x_i}{y} \frac{\partial x_i}{\partial y} \frac{y}{x_i} \tag{3}$$

where y is income. In Barten [60],

$$\overline{\omega}_{it} = \frac{\omega_{it} + \omega_{it-1}}{2} \tag{4}$$

where $\pi_{ij}$ is the Slutsky coefficient or substitution effect (the effect of a change in the price of good j on the demand for i, with income remaining constant):

$$\pi_{ij} = v_{ij} - \phi\theta_i\theta_j \tag{5}$$

where

$$\sum_j v_{ij} = \phi\theta_i = \frac{\partial y}{\partial \lambda} \frac{\lambda}{y} \frac{p_i x_i}{y} \frac{\partial x_i}{\partial y} \frac{y}{x_i} = \frac{\lambda p_i p_j u_{ij}}{y} \tag{6}$$

and $\lambda$ is the Lagrange coefficient.

We then have a system of multiple linear equations that is easy to estimate econometrically in relation to consumer theory. The dependent variables are the changes in the number of passengers using train transport and air transport and the synthetic index of car and bus passengers. This change in demand is explained by changes in real income and in the prices of the different modes of transport. Parameterisation implies that the coefficients $\theta_i$ and $\pi_{ij}$ are considered to be constants, and infinitesimal changes are taken as finite differences from t-1 to t. The overall impact of other important factors that influence the demand for passenger transport beyond prices and income, such as time cost, travel purpose, and personal driving preferences, are assumed to be random (Deaton [61]).

This study is based on the relevance of compliance with the hypothesis of the utility-maximising rational consumer theory of consumption. If this hypothesis is not fulfilled, then the demand models follow neither second-order nor first-order theoretical conditions (Serletis and Shahmoradi [62]). In that case, we cannot be sure that the demand for transport mode is a result of rational consumer decisions but must consider the possibility that this demand could be arbitrary, making it impossible to estimate meaningful choices. The theoretical restrictions are as follows. The sum of the change in expenditure for the different goods must be equal to the change in total income (Barten [60]):

$$\sum_{i=1}^{n} \theta_i = 1 \tag{7}$$

Moreover, in terms of zero-degree homogeneity, multiplying prices and income in the same proportion does not change demand decisions.

$$\sum_{j=1}^{n} \pi_{ij} = 0 \tag{8}$$

Third, regarding symmetry, the cross-derivatives of the Hicks demand function are equal:

$$\pi_{ij} = \pi_{ji} \tag{9}$$

The underlying idea is that if real income rises, consumers increase their consumption to match this additional income. However, if prices and nominal income increase by the same amount, consumption remains unchanged (there is no monetary illusion). Symmetry

is, however, not intuitive but rather a mathematical consequence of satisfying utility maximisation theory. Moreover, the expenditure function is concave. The Slutsky matrix [πij] is negative semidefinite, and the compensated own-price effect must be nonpositive:

$$H = \frac{\partial^2 C(u)}{\partial p_i \partial p_j} \leq 0 \tag{10}$$

$$\frac{\partial h_i(u, p)}{\partial p_i} \leq 0 \tag{11}$$

Therefore, all the eigenvalues of the [$\pi_{ij}$] matrix are less than or equal to 0. If the price of a mode of transport rises, abstracting from the impact on real income, it would be expected that consumption of that transport mode would decrease. The other values of the Slutsky matrix, cross-prices, must be positive. Again, abstracting from the effect on real income, consumers increase their consumption level of a particular good or service when the alternative product rises in price.

Finally, in terms of positivity, the cost function, which expresses the minimisation of consumption expenditure necessary to achieve a given level of utility, u, and given market prices, p, C(p,u), is nonnegative; for monotonicity, the derivatives of the cost function are nonnegative, and thus expenditure must increase monotonically with increasing prices. Aggregation, homogeneity, and symmetry are all set as conditions. Positivity and monotonicity must be observed. Negativity and curvature must be demonstrated from the [πij] matrix values.

The RDM preserves the conditions of the individual rational consumer at the aggregate level: "The convergence approach reveals that all of the restrictions at the individual consumer level carry over to the aggregate equations. This is the justification for applying the Rotterdam model to aggregate data" (Clements and Selvanathan [63], p. 64).

### 2.1.3. The Conditional Level of Demand Estimation

In this study, a conditional estimation is developed. Following the lead of Selvanathan and Selvanathan [64], the unconditional estimation of demand uses the shares of each good or service, or groups of them, to estimate total expenditure. The conditional estimation uses the expenditure allocation in the transport group as the group variable itself. That is, this paper's conditional estimation is specific to the details of the consumption group analysed (the demand for transport services on domestic routes in the United States). It assumes that consumer decisions regarding these services are made independently of those made for all other consumption groups (the block independence assumption). The reason for using this level of demand is that its estimate is based on the number of passengers using each mode of transport on each route rather than on the overall monetary expenditure at the national level.

Moreover, if the two-stage budget allocation assumption holds, then the allocation of expenditure is made first to the group (intercity transport) and then among the transport modes. Then, the conditional and unconditional demand functions should provide the same result (Edgerton [65]). In this sense, this paper follows the suggestion made by Clements and Selvananthan [63]: "There is a two-level decision hierarchy under block independence. In many applications such a separation into levels of the consumer's decision problem makes good intuitive sense" (p. 17). Finally, Muhammad and Countryman [55] recall one of the relevant advantages when choosing the RDM for estimation at the conditional level: "As noted by Theil and Clements (1987), blockwise dependence (or weak separability of product groups) is sufficient for limiting the analysis to goods within a product group when using the Rotterdam model" (p. 744).

The base equation used in Estimation (12) represents the conditional estimation of demand. We used the equation of Selvanathan and Selvanathan [64]:

$$\overline{w}_{it} Dq_{it} = \theta_i' \overline{\omega}_{gt} DQ_t + \sum_{j=1}^{n} \pi_{ij}^g Dp_{jt} + \epsilon_{it}^g \tag{12}$$

All terms on the right-hand side of the equation are expressed in terms of the i-th group of transport modes (Clements and Johnson [58]), where i denotes goods in Group g (referring to the group of goods and services consumed, which are modes of transport in this case); $\omega$ refers to the unconditional budget shares; and $\omega'$ refers to the conditional budget shares (which explain the allocation of expenditures in the transport group in terms of the group variable itself). $\overline{\omega}_{it}$ is defined in Equation (4), and the differentials are defined in Equation (1) and the following paragraph. The conditional marginal share of good i ($\theta'$) and the conditional Slutsky coefficients ($\pi$g) are then estimated. The terms are defined in the previous equations. The differences from the unconditional estimation are as follows:

$$\theta'_i = \frac{\theta_i}{\theta_g} \tag{13}$$

$$\theta_g = \sum_{i=1}^{n} \theta_i \tag{14}$$

$$DQ_{gt} = \sum_{i=1}^{n} \overline{\omega}'_{it} Dq_{it} \tag{15}$$

The restrictions remain the same.

Now, the conditional elasticities are as follows (Selvanathan and Selvanathan [64]):

$$\eta'_{iy} = \frac{\theta'_i}{\overline{\omega}'_{it}} \tag{16}$$

$$\eta'_{it} = \frac{\pi^g_{ij}}{\overline{\omega}_i} \tag{17}$$

The relationship between the Hicksian and Marshallian ($\in$ij) elasticities is given by the following:

$$\in_{ij} = \eta_{ij}{}^* + \omega_i \eta_{iy} \tag{18}$$

where the Marshall price elasticity is estimated by the usual relationship but now takes the conditional share of expenditure. The Marshall elasticity is equal to the Hicks elasticity plus the sum of the product of the share of the good and the income elasticity.

2.1.4. The Seemingly Unrelated Regression (SUR)

Finally, the assumption of the rational consumer hypothesis implies the existence of a series of multi-equational restrictions, which have already been discussed above. The imposition of these conditions requires the simultaneous estimation of all equations, which is conditional on these restrictions. For this reason, the seemingly unrelated regression method (SUR) introduced by Zellner [66] is applied. However, the application of RDM implies two important observations in this regard. On the one hand, the regressors on the right-hand side of the equation are equal and exogenous. The demand for all modes of transport is posed in terms of the same variables, i.e., the price differences between all modes of transport and real income. Thus, there is no covariance between the error terms of the model equations, meaning that the SUR estimation would not be more efficient than the estimation obtained by OLS. However, on the other hand, the number of global equations of the RDM system is larger than the number of independent variables. Therefore, the covariance matrix of the system would be singular. To solve this problem, one equation is dropped following the proposal of Barten [60], according to which any equation can be suppressed without affecting the results. Finally, restrictions are applied to the coefficients in order to estimate the omitted equation.

*2.2. Data*

This paper analyses 20 major passenger routes, with 9 of them exhibiting competition between the three modes of passenger transport (air, rail, and road) and 11 of them using

two modes (air and road). Data were extracted from 2003 to 2019. The routes were selected based on their importance, as measured by the number of passengers carried (the list of the analysed routes can be seen in Appendix A, Table A1). Data were obtained from different sources:

- Data on the percentage share of consumer expenditure were obtained from the Consumer Expenditure Surveys of the Bureau of Labor Statistics [67]. This included the following data: intercity bus fares, intercity train fares, airline fares, and car costs (intercity).
- Air passenger transport data were obtained from the US Bureau of Transportation Statistics, BTS, from the Airline Origin and Destination Survey [68]. These are random monthly surveys that consider 10% of the prices and quantities of all tickets sold in the United States. The number of passengers (daily average per quarter) and average fare per quarter were later annualised.
- Rail transport data were obtained from Amtrak (media, annual reports, the General Legislative Annual Report, and the Rail Passengers Association) [69–72], and they included a series of numbers reflecting passenger volume and average annual fare.
- Regarding road transport mode, there were no data on the number of passengers or the prices of cars and buses (Schwieterman et al. [73]). Therefore, these data had to be calculated on the basis of vehicle traffic, vehicle types, average number of passengers, miles, and unit fuel costs. Vehicle traffic counts were taken from [74–90].

First, the calculation of the number of passengers using each of the routes analysed, per car or bus, is calculated using Equations (19)–(21):

$$CTP = AADT \times 365 \times \%C \times VPC \tag{19}$$

$$BTP = AADT \times 365 \times \%B \times VPB \tag{20}$$

$$TP = CTP + BTP \tag{21}$$

Car total passengers (CTP) and bus total passengers (BTP) are calculated. The annual average daily traffic (AADT) [74–90], annualised over 365 days, is multiplied by the total traffic percentages, including cars and buses. BTS [1,2,91] provides data on the share of cars and buses in the overall traffic of the United States (cars, %C; bus, %B). Finally, it is necessary to multiply these numbers by the average number of passengers using both modes (cars VPC, buses VPB) on the basis of data taken from the FHWA [92,93]. The total value of passengers (TP) is the sum of both.

Road transport prices are obtained using two procedures that yield similar results. The first procedure is found in Equations (22)–(25).

$$AP = GASP \times \%GC + DIEP \times \%DC \tag{22}$$

$$CCOST = AP \times \frac{ROUD}{CMPG} / VPC \tag{23}$$

$$BCOST = DIEP \times \frac{ROUD}{BMPG} / VPB \tag{24}$$

where AP is the average fuel price, GASP is the average gasoline price (per gallon), and DIEP is the average diesel price (per gallon). Both values are multiplied by the percentage of cars using gasoline (%GC) and the percentage using diesel (%DC) [1,2]. Data were obtained from the EIA [94]. ROUD is the distance of the route in miles, which is divided by the miles per gallon to obtain the number of gallons required to travel the route (CMPG for cars, BMPG for buses). Fuel efficiency data were obtained from [1,2]. The prices per gallon of fuel per distance travelled, therefore, can be used to yield the cost per car (CCOST) and

the cost per bus (BCOST) of the route. This is performed by dividing these values by the number of passengers in each mode. Taking the weighted average of both costs relative to the number of passengers, the route price is found for road transport in (25).

$$PCB = \frac{CCOST \times CTP + BCOST \times BTP}{TP} \tag{25}$$

The second equivalent mode is based on statistics from [91–93] and the American Automobile Association [95,96]. This mode uses different costs (average cost of owning and operating an automobile). If we use the operating costs (fuel, maintenance, tires) of car travel, we arrive at results similar to those already obtained. If we include higher pro-rata ownership costs per mile, then the price of travel by car increases. It is reasonable to assume that the consumer considers the expenses directly associated with the trip to be the cost of the trip. The car is a durable consumer good, the usefulness of which cannot be associated with travelling from one end of a route to the other.

An example of the data obtained for two of the routes can be seen in Tables A2 and A3.

## 3. Results

After estimating the elasticities of the 20 routes considered in the study, we can affirm that 19 of them have coefficient estimates whose sign agrees with the theory's prediction. The one exception is the Portland–Seattle route, which has signs of estimated coefficients contrary to the theory's predictions (so that the Slutsky matrix is not negative semidefinite). The explanation for this seems to lie in a series of events that make the series non-homogeneous, with exogenous impacts on supply, masking the impact of prices and rents on demand (see Appendix B, Table A5).

Moreover, almost all estimated income elasticities were significant (for a maximum $p$ value of 0.1) for road and air transport modes, but not for rail transport. A total of 40 estimates of income elasticity were significant (38 without the Portland–Seattle route), 11 (10 without the Portland–Seattle route) were not statistically significant and, except for 2 of them, all referred to passenger rail transport.

Regarding the price elasticities of demand, approximately three-quarters of the coefficients were statistically significant when estimating demand for two modes of transport (air versus car/bus), but this significance decreased for estimates of routes with competition among all three modes of transport. On routes with three modes, about 45% of the price elasticities were statistically significant. The price elasticities of demand involving the rail mode of transport were generally not statistically significant. Finally, the application of the corresponding statistical tests, whose results are included in Appendix B at the end of Tables A4 and A5, confirmed that the vast majority of routes supported the validity of the regression hypotheses (no autocorrelation, homoscedasticity, normality, or relevance of the multi-equation conditions imposed).

The estimated values of the elasticities are shown in the following tables (Tables 1–3). (Appendix B shows Tables A4 and A5, which contain the estimated coefficients and statistical values).

Regarding air and road transport, the estimated average income elasticities show that air travel has an inelastic demand (0.94), while that of road travel is elastic (1.11). The coefficient of variation indicates the existence of some heterogeneity. As such, the median of each distribution was also calculated (0.95 air, 1.06 road), confirming the conclusions drawn from the average. The estimates made for rail transport were highly heterogeneous (with a high coefficient of variation), including both positive and negative estimates. The average is 0.05 (the median being 0.20). These estimates can be compared with those given in Tables A6–A8 in Appendix C. The quantification obtained is within the ranges observed in the scientific literature, although the income elasticity of road transport in the scientific literature is sometimes lower and that of air transport higher. After analysing a sample of 22 income elasticities from the literature, Holmgren [30] pointed out that the income elasticity of public transport demand ranged from −0.82 to 1.18, with an average

of 0.17. Thus, on the one hand, looking at the values in the tables in Appendix C and in Holmgren's study, there is a large dispersion of estimates, which underlines the need for further research. On the other hand, the low estimate of the income elasticity of the train is justified by the fact that many customers buy a car when their annual income reaches a certain level, which reduces the demand for public transport (train and bus modes).

The Hicks elasticities were negative in terms of the price itself and positive for the price of the competing modes (net substitutes). However, the substitution effects were all weak (a 10% increase in the price itself would generate, on average, a drop in demand for this mode of transport of −2.4% for air transport, −1.3% for road transport, and −8.2% for train transport). The Hicks cross-price elasticities were also weak. The median of each distribution confirmed the conclusions drawn from the average.

**Table 1.** Conditional income elasticities (shaded if significant). Averages were calculated taking Orlando/Washington (3 modes) into account (the calculations by the SUR method were carried out by R language (R core team), Systemfit package (Henningsen and Hamann [97,98])).

| Routes (i) | Period | Train | Air | Road |
|---|---|---|---|---|
| Los Angeles/Phoenix | 2007 to 2019 | | 1.2700 | 0.7922 |
| San Francisco/Los Angeles | 2007 to 2019 | −0.7325 | 0.5022 | 1.7372 |
| Los Angeles/Sacramento | 2007 to 2019 | −0.5685 | 0.7113 | 1.4610 |
| Las Vegas/San Francisco | 2003 to 2017 | | 0.3123 | 1.4914 |
| Atlanta/Miami | 2003 to 2017 | | 0.9357 | 1.0445 |
| Chicago/Washington DC | 2008 to 2019 | 0.3071 | 0.6582 | 1.4803 |
| Atlanta/New York | 2003 to 2017 | | 1.2307 | 0.8406 |
| Chicago/New York | 2003 to 2019 | | 0.5832 | 1.2970 |
| San Francisco/Seattle | 2003 to 2017 | | 1.5766 | 0.5891 |
| Orlando/Washington (2 modes) | 2003 to 2017 | | 1.0800 | 0.9447 |
| Orlando/Washington (3 modes) | 2003 to 2017 | 0.3287 | 0.7386 | 1.3747 |
| Denver/Los Angeles | 2003 to 2019 | | 1.3833 | 0.7270 |
| Boston/Chicago | 2004 to 2019 | | 0.9603 | 1.0300 |
| New York/Orlando | 2003 to 2017 | | 1.0178 | 0.9875 |
| Chicago/Orlando | 2003 to 2017 | | 1.2322 | 0.8394 |
| Miami/Washington | 2008 to 2019 | 0.2036 | 0.6045 | 1.5555 |
| New York/Miami | 2003 to 2019 | | 0.7611 | 1.1713 |
| Chicago/St. Louis | 2007 to 2019 | 0.1728 | 0.4853 | 1.7197 |
| Buffalo/New York City | 2007 to 2019 | 0.3365 | 1.6900 | 0.1269 |
| Portland/Seattle | 2003 to 2019 | −0.0689 | 1.1823 | 0.8016 |
| Chicago/Detroit | 2003 to 2018 | 0.4799 | 0.9686 | 1.0722 |
| Estimated average elasticity | | 0.0510 | 0.9402 | 1.1070 |
| Estimated S.D. of elasticity | | 0.4270 | 0.3801 | 0.4124 |
| Coefficient of variation (%) | | 8.3771 | 0.4043 | 0.3726 |
| Estimated median | | 0.2036 | 0.9480 | 1.0584 |

[i] The Slutsky coefficient matrix must be negative semidefinite, i.e., no eigenvalue is positive. One of them must be 0 as a result of the imposed conditions. This concavity is checked for all routes. For example, the eigenvalues of the Chicago and St. Louis route are −0.0001 (approx. 0), −0.0005, and −0.0007.

The Marshall elasticities show inelastic demands for all modes of transport. A 10% increase in the price itself implies, on average, a reduction of −6.2% for air, −6.9% for road transport, and −8.2% for train transport. All cross-price elasticities are weak. These elasticities are sometimes negative because the income effects dominate the substitution effects (grossly complementary to one another) (income effects predominate for air transport demand relative to road transport prices, as well as for road transport relative to the prices of the other two modes (the strongest effect is the 6% fall in road transport demand due to the 10% increase in air transport prices)).

**Table 2.** Conditional Marshall price elasticities (shaded if significant).

| Routes | Period | Transport Mode | | | Cross Elasticity | |
|---|---|---|---|---|---|---|
| | | Train | Air | Road | Air Pass., Road Prices | Road Pass., Air Prices |
| Los Angeles/Phoenix | 2007 to 2019 | | −0.6597 | −0.5304 | −0.6104 | −0.2618 |
| San Francisco/Los Angeles | 2007 to 2019 | −1.6481 | −0.5921 | −0.8859 | 0.0486 | −0.7582 |
| Los Angeles/Sacramento | 2007 to 2019 | −0.2665 | −0.9204 | −1.0185 | 0.2186 | −0.4169 |
| Las Vegas/San Francisco | 2003 to 2017 | | −0.3866 | −1.0532 | 0.0743 | −0.4382 |
| Atlanta/Miami | 2003 to 2017 | | −0.5702 | −0.7466 | −0.3655 | −0.2979 |
| Chicago/Washington DC | 2008 to 2019 | −0.9325 | −0.4392 | −0.6524 | −0.2506 | −0.7956 |
| Atlanta/New York | 2003 to 2017 | | −0.6690 | −0.6111 | −0.5617 | −0.2296 |
| Chicago/New York | 2003 to 2019 | | −0.5655 | −0.9861 | −0.0177 | −0.3109 |
| San Francisco/Seattle | 2003 to 2017 | | −0.7784 | −0.4308 | −0.7982 | −0.1583 |
| Orlando/Washington (2 modes) | 2003 to 2017 | | −0.7024 | −0.7377 | −0.3776 | −0.2070 |
| Orlando/Washington (3 modes) | 2003 to 2017 | −1.0233 | −0.6360 | −0.6923 | −0.1595 | −0.6408 |
| Denver/Los Angeles | 2003 to 2019 | | −0.7503 | −0.5485 | −0.6330 | −0.1785 |
| Boston/Chicago | 2004 to 2019 | | −0.5153 | −0.6757 | −0.4450 | −0.3543 |
| New York/Orlando | 2003 to 2017 | | −0.6255 | −0.7277 | −0.3923 | −0.2598 |
| Chicago/Orlando | 2003 to 2017 | | −0.6764 | −0.6151 | −0.5558 | −0.2243 |
| Miami/Washington | 2008 to 2019 | −0.4137 | −0.4719 | −0.7517 | −0.1406 | −0.7700 |
| New York/Miami | 2003 to 2019 | | −0.4865 | −0.8042 | −0.2746 | −0.3671 |
| Chicago/St. Louis | 2007 to 2019 | −0.9741 | −0.3148 | −0.7901 | −0.1873 | −0.9420 |
| Buffalo/New York City | 2007 to 2019 | −0.6482 | −0.9692 | −0.0575 | −0.7108 | −0.0681 |
| Portland/Seattle | 2003 to 2019 | | | | | |
| Chicago/Detroit | 2003 to 2018 | −0.6596 | −0.8174 | −0.6218 | −0.1426 | −0.4399 |
| Estimated average Marshall elasticity (1) | | −0.8208 | −0.6234 | −0.6947 | −0.3107 | −0.4164 |
| Estimated S.D. of Marshall elasticity | | 0.4293 | 0.1728 | 0.2270 | 0.2873 | 0.2483 |
| Coefficient of variation | | 0.5231 | 0.2773 | 0.3268 | 0.9246 | 0.5962 |
| Estimated median | | −0.7961 | −0.6255 | −0.6923 | −0.2746 | −0.3543 |

(1) Averages were calculated taking Orlando/Washington (3 modes) into account.

**Table 3.** Conditional Marshall cross-price elasticities.

| Cross Elasticities | | Train Passengers | | Air Passengers | | Road Passengers | |
|---|---|---|---|---|---|---|---|
| Routes | Period | Air Prices | Road Prices | Train Prices | Road Prices | Train Prices | Air Prices |
| San Francisco/Los Angeles | 2007 to 2019 | 1.4061 | 0.9745 | 0.0412 | 0.0486 | −0.0931 | −0.7582 |
| Los Angeles/Sacramento | 2007 to 2019 | 0.4320 | 0.4030 | −0.0095 | 0.2186 | −0.0256 | −0.4169 |
| Chicago/Washington DC | 2008 to 2019 | 0.7159 | −0.0904 | 0.0317 | −0.2506 | −0.0323 | −0.7956 |
| Orlando/Washington (3 modes) | 2003 to 2017 | 1.2216 | −0.5270 | 0.0569 | −0.1595 | −0.0416 | −0.6408 |
| Miami/Washington | 2008 to 2019 | 0.2963 | −0.0862 | 0.0080 | −0.1406 | −0.0338 | −0.7700 |
| Chicago/St. Louis | 2007 to 2019 | 0.4303 | 0.3709 | 0.0167 | −0.1873 | −0.0201 | −0.9420 |
| Buffalo/New York City | 2007 to 2019 | 0.3326 | −0.0208 | −0.0100 | −0.7108 | −0.0014 | −0.0681 |
| Chicago/Detroit | 2003 to 2018 | −0.0464 | 0.2262 | −0.0087 | −0.1426 | −0.0105 | −0.4399 |
| Estimated average Marshall elasticity | | 0.5985 | 0.1563 | 0.0158 | −0.1655 | −0.0323 | −0.6039 |
| Estimated S.D. of Marshall elasticity | | 0.4911 | 0.4471 | 0.0255 | 0.2672 | 0.0278 | 0.2810 |
| Coefficient of variation | | 0.8205 | 2.8610 | 1.6164 | 1.6140 | 0.8611 | 0.4653 |
| Estimated median | | 0.4311 | 0.1027 | 0.0123 | −0.1510 | −0.0289 | −0.6995 |

For example, considering the route from Chicago to Orlando, a 10% increase in air transport prices causes a −6.8% reduction in the number of passengers using air transport and a −2.2% reduction in those using road transport. Moreover, a 10% increase in road transport prices causes a −5.6% reduction in the demand for air transport and a −6.2% reduction in the demand for passenger transport using cars or buses. Taking, for example, the Los Angeles to San Francisco route, a 10% increase in the price of rail would imply a −16.5% decrease in the use of rail, with almost imperceptible changes in the use of air (+0.4%) and cars or buses (−0.9%). The percentage change in the price of air travel would mean an increase in train use (14.1%) and a reduction in air travel (−5.9%) and car/bus use (−7.6%).

Different causes can be found for the nonstatistical significance of the coefficients calculated for rail transport and the variability of the estimated values. This is a problem that has been raised, in one form or another, in previous analyses. There are few studies of passenger rail elasticities in the United States. Some of the studies that have been performed estimate coefficients that are not statistically significant. Finally, there is a great deal of variability in the elasticities calculated. See Appendix C, Table A8. On the one hand, the demand for rail transport might be influenced by some factors not considered in the model. The conclusions of Wardman [99], who points out in his UK study that "the critical importance of GDP for rail demand growth is quite clear" (p. 15), may be relevant. On the other hand, the lack of rail demand data concerning seat types, tickets for journey types, etc., and the fact that the data sample is smaller in size make this value difficult to estimate. The complexity of the pricing strategy and the importance of detailed data on the prices paid by different demanders is highlighted in the study of Cirillo and Hetrakul [40], who stated that "Amtrak's pricing strategy is more complicated than the one presented in this paper. We have not taken into account cancellation behaviour, various discounts, guest reward programmes and special fare schemes" (p. 21). In general, data on and knowledge of Amtrak's fare policy need to be improved.

## 4. Discussion

Based on the objectives set out at the beginning of this research, the following results have been obtained. On the one hand, the method used and the way of obtaining the data and reconstructing the usable series have been shown to enable comparisons of the quantifications made for the different routes and conclusions to be reached for the routes as a whole and for each route individually. In this sense, the procedure makes it possible, in a relatively simple and rapid way, to relate the elasticities of demand thus estimated and the effectiveness of the different public policies to be applied to improve the sustainability of passenger transport in the United States on domestic routes using the data available from public statistical bodies. In summary, a method based on microeconomic theory and annually available data has been applied with relative success. On the other hand, the elasticities of demand for passenger transport on domestic routes in the United States have been estimated, although the estimate is not significant in relation to rail transport. Further research is needed on the explanatory factors of transport and the need for a more reliable understanding of the behaviour of rail passenger transport.

A statistically significant estimate of the elasticity of demand for air and road passenger transport has been achieved for the main US domestic routes: the average values calculated for the routes indicate that all income elasticities are positive (normal goods). While air transport demand is slightly inelastic and road transport demand is slightly elastic, both close to unity, rail demand is highly inelastic (although the estimate is not reliable due to the high coefficient of variation). This implies a crucial fact for planning future transport infrastructure and services in the United States: future income increases in the United States will lead to a roughly proportional increase in demand for air and road transport at the route level (somewhat higher for the latter mode). Although the exact value of the income elasticity of rail transport in the US has not been quantified with certainty, it should be reasonably low. It appears that demanders, given the current characteristics of rail

transport in the US, tend to increase their demand for this mode very little in response to rising incomes. Thus, demand pressure will be differential for each of the modes. However, it is possible to note, at least as an approximation, the apparent differences between the different routes.

All the Hicksian elasticities show consistent signs of being competing services, with all the modes of transport being net substitutes. However, these elasticities are very low, and the price substitution effects are relatively weak. Moreover, all modes of transport have negative Marshallian demand own-price elasticities, although their values are less than one. Increases in transportation prices or costs produce a less-than-proportional reduction in the use of this service. This means that demanders have little propensity to stop using a transport type or to switch between transport modes. That is, in the face of price movements and differential price changes between modes, the demander has a high propensity to continue to use a particular mode of transport on US domestic routes. This can be because "transport is a derived demand and tends to be inelastic" (Oum et al. [28], p. 8). Moreover, the Marshallian demand for air and car/bus transport shows that the two are strongly complementary to each other. This is because the income effect has a much stronger impact than the substitution effect; that is, the change in real income resulting from a change in prices has a greater influence than the direct impact of prices on switching between modes.

In general, the proportionally smaller demand response to price increases, and the reduced or non-existent transmission of demand from one mode to another at the route level (as shown by the values of the cross-price elasticities), indicate that fiscal or carbon price actions require very high price increases to produce significant reductions in transport demand, which are always proportionally smaller than the price increase and, consequently, entail high costs for consumers. While the fact of the inelastic price elasticity of demand for energy implies as an additional effect that a tax on $CO_2$ production would strongly increase tax revenues (Halsnaen et al. [100], p. 153), it should be asked, in the face of high increases in transportation costs, what social and political impact this would have. Barrett and Chen indicate that "prices, particularly of food and fuel, seem to be particularly important" for explaining recent social unrest [101]. This would indicate that public policy interventions through regulation would be more successful in the interests of sustainability (legal changes to the types of vehicles or types of fuel to be used). The most notable measure of this kind is that from the State of California, which has banned new gas-powered cars by 2035 (Emma Newburger [102]). Nakamura and Hayashi [103] argue that the introduction of a price on $CO_2$ output would reduce emissions by 5%, while the fuel efficiency improvement is estimated to be more than 20% (although it could generate the opposite effect of increasing travel demand and traffic congestion). Major investments in the infrastructure of different modes are also possible at the route level, which would change either the time taken from one end of the route to the other by each mode, or the possibility of interchanging one mode for another (e.g., rail terminals in the vicinity of airports). Major changes in infrastructure would also change the demand for each mode.

Moreover, since increased income generates greater proportional increases in the demand for road transport, investment policies for this mode are needed in anticipation of an expanding economic cycle to avoid infrastructure congestion. Similarly, as the substitution effect between modes is very weak, measures should be taken to increase the exchange of demand between modes. The direct interconnection of airports by rail and via the routes offered by Amtrak seems to be a crucial element. Such a connection already exists, for example, at the Baltimore/Washington International Airport station, and allows travellers to arrive or depart on Amtrak's Northeast Corridor routes (Amtrak [69]). In addition, price elasticity is also key in estimating the impact of changes in fuel prices based on oil price fluctuations.

Another interesting result of this study is the existence of a certain heterogeneity in the distribution of the estimated demand elasticities. This necessarily requires the study of the underlying and differential factors of each route, as well as the design of efficient public

transport policies for each route. However, any effective public policy should quantify costs and benefits, which requires a view of all transportation routes as a whole.

Overall, therefore, a strategy for reducing greenhouse gas emissions into the environment should seek to replace fossil fuels with cleaner modes of energy, especially for those modes that will see the greatest increase in demand: planes, cars, and buses. This should primarily be performed with electric engines using electricity produced from renewable sources or sustainable aviation fuel (OEE&RE [104]), or otherwise using equivalents for vehicle engines.

This research could be extended in several directions in future research. The first and most immediate objective in this field is to confirm the findings of this research by analysing domestic routes in other locations and expanding the sample size to achieve greater precision. In particular, it is necessary to further estimate the demand for intercity trains in the United States and, in general, in all countries of the world. In addition, the geographical expansion of the study would benefit from extending the research to international transport routes. On the other hand, constructing and estimating a dynamic demand model that considers the possible impact of time would also be of great interest. Moreover, incorporating additional factors related to consumer preferences into the model would be of great value.

**Author Contributions:** Conceptualisation, I.E.R. and M.T.-J.; methodology, I.E.R.; software, I.E.R.; validation, I.E.R. and M.T.-J.; formal analysis, I.E.R.; investigation, I.E.R.; data curation, I.E.R.; writing—original draft preparation, I.E.R. and M.T.-J.; writing—review and editing, I.E.R. and M.T.-J.; visualisation, I.E.R. and M.T.-J.; supervision, I.E.R., M.T.-J. and M.C.-R.; project administration, I.E.R., M.T.-J. and M.C.-R.; funding acquisition, M.T.-J. and M.C.-R. All authors have read and agreed to the published version of the manuscript.

**Funding:** This work has been partially subsidised by the "Ministerio de Ciencia e Innovación (España)" (grant reference: PID2020-115454GB-C22) and the "Consejería de Economía, Conocimiento, Empresas y Universidad (Junta de Andalucía)" (grant reference: P20_00074).

**Data Availability Statement:** Data supporting the reported results can be found; references have been included.

**Conflicts of Interest:** The authors declare no conflict of interest.

**Appendix A.**

**Table A1.** Selection of 20 routes. Data for 2017, quarter 2. Data: BTS (2023a) [68].

| CITY 1 | CITY 2 | Non-Stop Market Miles (Using Radian Measure) | Air Passengers Per Day |
|---|---|---|---|
| Miami, FL (Metropolitan Area) | New York City, NY (Metropolitan Area) | 1139 | 16,799 |
| Chicago, IL | Orlando, FL | 1005 | 4695 |
| New York City, NY (Metropolitan Area) | Orlando, FL | 989 | 11,317 |
| Miami, FL (Metropolitan Area) | Washington, DC (Metropolitan Area) | 946 | 6474 |
| Boston, MA (Metropolitan Area) | Chicago, IL | 867 | 5148 |
| Denver, CO | Los Angeles, CA (Metropolitan Area) | 862 | 5853 |
| Atlanta, GA (Metropolitan Area) | New York City, NY (Metropolitan Area) | 795 | 7759 |

**Table A1.** *Cont.*

| CITY 1 | CITY 2 | Non-Stop Market Miles (Using Radian Measure) | Air Passengers Per Day |
|---|---|---|---|
| Orlando, FL | Washington, DC (Metropolitan Area) | 787 | 5620 |
| Chicago, IL | New York City, NY (Metropolitan Area) | 773 | 12,372 |
| San Francisco, CA (Metropolitan Area) | Seattle, WA | 696 | 6857 |
| Chicago, IL | Washington, DC (Metropolitan Area) | 622 | 6306 |
| Atlanta, GA (Metropolitan Area) | Miami, FL (Metropolitan Area) | 594 | 4725 |
| Las Vegas, NV | San Francisco, CA (Metropolitan Area) | 414 | 7011 |
| Los Angeles, CA (Metropolitan Area) | Sacramento, CA | 404 | 5709 |
| Los Angeles, CA (Metropolitan Area) | San Francisco, CA (Metropolitan Area) | 372 | 22,488 |
| Los Angeles, CA (Metropolitan Area) | Phoenix, AZ | 370 | 4975 |
| Buffalo, NY | New York City, NY (Metropolitan Area) (Maple, Empire and Lake Routes) | 326 | 1697 |
| Chicago, IL | St. Louis, MO (Lincoln Route) | 258 | 1193 |
| Chicago, IL | Detroit, MI (Wolverine Route) | 235 | 1400 |
| Portland, OR | Seattle, WA (Cascades Route) | 129 | 819 |

**Table A2.** Quantity and price data for an example route (Chicago to New York City), in log-changes, Dqit = lnqit – lnqit − 1 y Dpit = lnpit – lnpit − 1, where i = 1, 2.

| Year | Number of Passengers | | Prices | |
|---|---|---|---|---|
| | Air | Car/Bus | Air | Car/Bus |
| 2001 | −0.1003 | | −0.1140 | 0.1044 |
| 2002 | 0.0631 | 0.3878 | −0.2229 | −0.1188 |
| 2003 | −0.0206 | 0.0351 | 0.0081 | 0.0787 |
| 2004 | 0.0872 | 0.0027 | −0.0189 | 0.0014 |
| 2005 | 0.0476 | 0.1563 | −0.0597 | 0.0734 |
| 2006 | 0.1074 | −0.0299 | −0.0476 | 0.0693 |
| 2007 | 0.0356 | 0.0000 | −0.0384 | 0.0036 |
| 2008 | −0.1385 | 0.0416 | 0.2470 | 0.1208 |
| 2009 | −0.0562 | −0.1511 | −0.1547 | −0.1171 |
| 2010 | 0.0983 | 0.0223 | 0.0489 | 0.1225 |
| 2011 | 0.0378 | 0.0102 | 0.0289 | 0.0531 |

**Table A2.** *Cont.*

| Year | Number of Passengers | | Prices | |
|---|---|---|---|---|
| | Air | Car/Bus | Air | Car/Bus |
| 2012 | 0.0232 | −0.0967 | 0.0430 | 0.0954 |
| 2013 | −0.0151 | −0.0219 | 0.1064 | 0.0089 |
| 2014 | −0.0056 | 0.0065 | 0.0706 | −0.0518 |
| 2015 | 0.0866 | 0.1093 | −0.1598 | −0.1045 |
| 2016 | 0.0779 | −0.0692 | −0.1021 | −0.1137 |
| 2017 | 0.0336 | −0.0748 | −0.0236 | 0.1716 |
| 2018 | 0.0034 | −0.0017 | 0.0760 | 0.0498 |
| 2019 | −0.0165 | −0.0862 | 0.0390 | 0.1551 |

**Table A3.** Divisia indexes, Miami to Washington. Volume: $DQ_t = \sum_{i=1}^{3} \overline{w}'_t Dq_{it}$.

| Year | Rail | Air | Car/bus | Sum (DQgt) |
|---|---|---|---|---|
| 2001 | – | −0.0145 | −0.0163 | – |
| 2002 | – | 0.0070 | 0.1265 | – |
| 2003 | – | 0.0264 | 0.0099 | – |
| 2004 | – | 0.0494 | 0.0018 | – |
| 2005 | – | 0.0218 | −0.0373 | – |
| 2006 | – | −0.0020 | 0.0094 | – |
| 2007 | – | 0.0028 | −0.0044 | – |
| 2008 | 0.0019 | −0.0227 | 0.0198 | −0.0010 |
| 2009 | −0.0002 | −0.0034 | 0.0560 | 0.0525 |
| 2010 | 0.0009 | 0.0081 | −0.0111 | −0.0021 |
| 2011 | 0.0012 | 0.0346 | −0.0137 | 0.0222 |
| 2012 | 0.0003 | 0.0064 | 0.0156 | 0.0224 |
| 2013 | 0.0001 | 0.0196 | −0.0133 | 0.0064 |
| 2014 | 0.0007 | −0.0091 | 0.0248 | 0.0165 |
| 2015 | −0.0002 | 0.0265 | 0.0327 | 0.0590 |
| 2016 | −0.0022 | 0.0141 | 0.0169 | 0.0287 |
| 2017 | −0.0008 | 0.0185 | 0.0130 | 0.0307 |
| 2018 | −0.0005 | −0.0168 | −0.0145 | −0.0319 |
| 2019 | 0.0013 | −0.0215 | −0.0727 | −0.0928 |

## Appendix B. Detailed Results

Statistical tests are applied to detect four possible problems: autocorrelation, heteroscedasticity, or non-normality of the regression residuals (Breusch–Godfrey test [105,106], Breusch–Pagan test [107], Anderson–Darling normality test [108]), as well as the non-statistical significance of the multiple equation conditions (likelihood ratio test [109] and Theil's F-test [110]) (using language R version 4.2.3, packages lmtest, nortest [97,111,112]). The problems encountered are indicated at the end of the tables.

**Table A4.** Estimated conditional coefficients. Equation (12). The first 10 routes studied.

| Routes | | | Car Fares: Gasoline Cost. Bus Fares: Diesel Cost | | | | Average (t, t−1) Unconditional/ Conditional Budget Shares |
| --- | --- | --- | --- | --- | --- | --- | --- |
| | | | Marginal Budget Share | Slutsky Coefficients | Slutsky Coefficients | Slutsky Coefficients | |
| Route. Period. | Transport Mode | Values | $\theta i$ | $\pi i1$ Train | $\pi i2$ Air | $\pi i3$ Car/Bus | $\varpi it$ |
| Los Angeles/ Phoenix | Air (i = 1). | Coefficients | 0.5523 | | −0.0008 | 0.0008 | 0.0071 |
| 2007 to 2019 | | Standard errors(SE) | 0.124 | | 0.0005 | 0.0005 | 0.4349 |
| | | Pr(>\|t\|) | 0.0012 | | 0.1407 | 0.1407 | |
| | Car/bus (i = 2). | Coeff. | 0.4477 | | 0.0008 | −0.0008 | 0.0093 |
| | | SE | 0.124 | | 0.0005 | 0.0005 | 0.5651 |
| | | Pr(>\|t\|) | 0.0048 | | 0.1407 | 0.1407 | |
| San Francisco (Oakland)/Los Angeles (1) | Train (i = 1) | Coeff. | −0.0162 | −0.0006 | 0.0004 | 0.0002 | 0.0004 |
| 2007 to 2017 | | SE | 0.0328 | 0.0004 | 0.0003 | 0.0004 | 0.0221 |
| | | Pr(>\|t\|) | 0.6269 | 0.0999 | 0.1728 | 0.5511 | |
| | Air (i = 2). | Coeff. | 0.2776 | 0.0004 | −0.0022 | 0.0019 | 0.0071 |
| | | SE | 0.0592 | 0.0003 | 0.0005 | 0.0005 | 0.4252 |
| | | Pr(>\|t\|) | 0.0002 | 0.1728 | 0.0078 | 0.0014 | |
| | Car/bus (i = 3). | Coeff. | 0.7386 | −0.0005 | 0.0019 | −0.0014 | 0.0093 |
| | | SE | 0.0548 | 0.0003 | 0.0005 | 0.0005 | 0.5527 |
| | | Pr(>\|t\|) | 0 | 0.126 | 0.0014 | 0.0131 | |
| Los Angeles/ Sacramento (2) | Train (i = 1) | Coeff. | −0.0124 | −0.0001 | 0 | 0.0001 | 0.0004 |
| 2007 to 2019 | | SE | 0.0205 | 0.0002 | 0.0002 | 0.0002 | 0.0219 |
| | | Pr(>\|t\|) | 0.5507 | 0.6253 | 0.8197 | 0.7911 | |
| | Air (i = 2). | Coeff. | 0.3952 | 0 | −0.0037 | 0.0037 | 0.0071 |
| | | SE | 0.0982 | 0.0002 | 0.0011 | 0.001 | 0.4225 |
| | | Pr(>\|t\|) | 0.0007 | 0.8197 | 0.0026 | 0.0017 | |
| | Car/bus (i = 3). | Coeff. | 0.6172 | 0.0001 | 0.0037 | −0.0038 | 0.0093 |
| | | SE | 0.0928 | 0.0002 | 0.001 | 0.001 | 0.5556 |
| | | Pr(>\|t\|) | 0 | 0.7911 | 0.0017 | 0.0009 | |
| Las Vegas/ San Francisco | Air (i = 1). | Coeff. | 0.1299 | | −0.0018 | 0.0018 | 0.007 |
| 2003 to 2019 | | SE | 0.0575 | | 0.0008 | 0.0008 | 0.416 |
| | | Pr(>\|t\|) | 0.0404 | | 0.0445 | 0.0445 | |
| | Car/bus (i = 2). | Coeff. | 0.871 | | 0.0018 | −0.0018 | 0.0098 |
| | | SE | 0.0576 | | 0.0008 | 0.0008 | 0.584 |
| | | Pr(>\|t\|) | 0 | | 0.0445 | 0.0445 | |
| Atlanta/Miami | Air (i = 1). | Coeff. | 0.3823 | | −0.0013 | 0.0013 | 0.0068 |
| 2003 to 2017 | | SE | 0.0877 | | 0.0008 | 0.0008 | 0.4086 |
| | | Pr(>\|t\|) | 0.0009 | | 0.1253 | 0.1253 | |
| | Car/bus (i = 2). | Coeff. | 0.6177 | | 0.0013 | −0.0013 | 0.0099 |
| | | SE | 0.0877 | | 0.0008 | 0.0008 | 0.5914 |
| | | Pr(>\|t\|) | 0 | | 0.1253 | 0.1253 | |
| Chicago/ Washington DC (3) | Train (i = 1) | Coeff. | 0.0068 | −0.0003 | 0.0003 | 0 | 0.0004 |
| 2007 to 2019 | | SE | 0.0348 | 0.0005 | 0.0005 | 0.0002 | 0.0221 |
| | | Pr(>\|t\|) | 0.8473 | 0.4638 | 0.4786 | 0.9299 | |
| | Air (i = 2). | Coeff. | 0.3638 | 0.0003 | −0.0005 | 0.0002 | 0.0071 |
| | | SE | 0.1417 | 0.0005 | 0.0008 | 0.0008 | 0.4252 |
| | | Pr(>\|t\|) | 0.0181 | 0.4786 | 0.5502 | 0.7846 | |
| | Car/bus (i = 3). | Coeff. | 0.6294 | 0 | 0.0002 | −0.0002 | 0.0093 |
| | | SE | 0.1456 | 0.0001 | 0.0008 | 0.0007 | 0.5527 |
| | | Pr(>\|t\|) | 0.0004 | 0.9668 | 0.7846 | 0.7702 | |

**Table A4.** *Cont.*

| Routes | | | Car Fares: Gasoline Cost. Bus Fares: Diesel Cost | | | | Average (t, t−1) Unconditional/ Conditional Budget Shares |
|---|---|---|---|---|---|---|---|
| | | | Marginal Budget Share | Slutsky Coefficients | Slutsky Coefficients | Slutsky Coefficients | |
| Route. Period. | Transport Mode | Values | θi | πi1 Train | πi2 Air | πi3 Car/Bus | ϖit |
| Atlanta/New York | Air (i = 1). | Coeff. | 0.5028 | | −0.0011 | 0.0011 | 0.0068 |
| 2003 to 2017 | | SE | 0.1023 | | 0.0005 | 0.0005 | 0.4086 |
| | | Pr(>|t|) | 0.0004 | | 0.0381 | 0.0381 | |
| | Car/bus (i = 2). | Coeff. | 0.4972 | | 0.0011 | −0.0011 | 0.0099 |
| | | SE | 0.1023 | | 0.0005 | 0.0005 | 0.5914 |
| | | Pr(>|t|) | 0.0004 | | 0.0381 | 0.0381 | |
| Chicago/New York | Train (i = 1) | Coeff. | 0.2426 | | −0.0022 | 0.0022 | 0.007 |
| 2003 to 2019 | | SE | 0.0906 | | 0.0009 | 0.0009 | 0.416 |
| | | Pr(>|t|) | 0.018 | | 0.0245 | 0.0245 | |
| | Air (i = 2). | Coeff. | 0.7574 | | 0.0022 | −0.0022 | 0.0098 |
| | | SE | 0.0906 | | 0.0009 | 0.0009 | 0.584 |
| | | Pr(>|t|) | 0 | | 0.0244 | 0.0244 | |
| San Francisco/ Seattle (4) | Air (i = 1). | Coeff. | 0.6559 | | −0.0009 | 0.0009 | 0.007 |
| 2003 to 2017 | | SE | 0.1403 | | 0.0005 | 0.0005 | 0.416 |
| | | Pr(>|t|) | 0.0004 | | 0.1125 | 0.1125 | |
| | Car/bus (i = 2). | Coeff. | 0.344 | | 0.0009 | −0.0009 | 0.0098 |
| | | SE | 0.1403 | | 0.0005 | 0.0005 | 0.584 |
| | | Pr(>|t|) | 0.0279 | | 0.1125 | 0.1125 | |
| Orlando/ Washington (5) | Train (i = 1) | Coeff. | 0.0073 | −0.0004 | 0.0005 | −0.0001 | 0.0004 |
| 2003 to 2019 | | SE | 0.0403 | 0.0003 | 0.0004 | 0.0002 | 0.0221 |
| | | Pr(>|t|) | 0.8589 | 0.2398 | 0.2076 | 0.476 | |
| | Air (i = 2). | Coeff. | 0.4082 | 0.0005 | −0.0016 | 0.0011 | 0.0071 |
| | | SE | 0.1072 | 0.0004 | 0.0008 | 0.0007 | 0.4252 |
| | | Pr(>|t|) | 0.0014 | 0.2076 | 0.0482 | 0.1244 | |
| | Car/bus (i = 3). | Coeff. | 0.5845 | −0.0001 | 0.0011 | −0.001 | 0.0093 |
| | | SE | 0.1096 | 0.0002 | 0.0007 | 0.0007 | 0.5527 |
| | | Pr(>|t|) | 0 | 0.5843 | 0.1244 | 0.1469 | |

(1) Approximately 10.5% of passengers use this on the Oakland (SF)–Los Angeles route (Amtrak Ridership Statistics [72]). This relative irrelevance, about 45 thousand passengers per train with more than 8 million air passengers in 2017, would justify the non-statistical significance of the coefficients associated with the train. In addition, several of the regression assumptions (non-normality, autocorrelation of the residuals, and non-relevance of the multi-equation restrictions) are not met. (2) On this route, LA–Sacr., car transport prices are used, including not only the cost of petrol but also the total cost of all items, including the cost of vehicle use and purchase, pro-rated by mileage. The regression results improve significantly with this option. (3) Chicago–Washington DC, a number of conditions are not met. Multi-equation restrictions significantly worsen the results: likelihood ratio test, Pr(>Chisq) = 0.0000, Theil's F-test: Pr(>F) 0.0051. Autocorrelation in the residuals of the train mode regression: Breusch–Godfrey test for serial correlation of order up to 1, *p* value = 0.0038. Homoscedasticity hypotheses are not satisfied in the car/bus mode regression: studentized Breusch–Pagan test, *p* value = 0.06083; the normality of the residuals is not in the road transport regression. Anderson–Darling normality test, *p* value = 0.0191. (4) Accidents and interruptions occur that do not allow access to a sufficiently long homogeneous series. For this reason, the train mode of transport is not included in the regression. The regression also fails the hypothesis of non-autocorrelation of residuals in the road transport equation in terms of meeting autocorrelation. Breusch–Godfrey test, *p* value = 0.07928. The same holds for the hypothesis of normality of these residuals: Anderson–Darling normality test, *p* value = 0.02037. (5) There is an Amtrak route called Auto Train (855 miles). By including the number of passengers and the average ticket, the regression does not meet the set conditions. This is plausibly due to the variety of prices and services offered by the route: sleepers or not, Auto Train car or not. See, for example, Sibdari et al. (2008) [113], where base cost and upgrade cost are distinguished. When disregarding the data for the train route, the regression is relevant and meets the conditions. This is probably because the maximum number of travellers taking the ends of the route does not exceed 10% of the travellers using air transport. For example, in 2018, about 2 million people travelled by air, while 225 thousand people were transported by train.

**Table A5.** Estimated conditional coefficients. Equation (12). The following 10 routes studied, plus Orlando/Washington (two modes).

| Routes | | | Car Fares: Gasoline Cost. Bus Fares: Diesel Cost. | | | | Average (t, t−1) Unconditional/ Conditional Budget Shares |
|---|---|---|---|---|---|---|---|
| | | | Marginal Budget Share | Slutsky Coefficients | Slutsky Coefficients | Slutsky Coefficients | |
| Route. Period. | Transport Mode | Values | θi | πi1 Train | πi2 Air | πi3 Car/Bus | ϖit |
| Orlando/ Washington (3) | Air (i = 1). | Coeff. | 0.4413 | | −0.0018 | 0.0018 | 0.0068 |
| 2003 to 2019 | | SE | 0.0858 | | 0.0009 | 0.0009 | 0.4086 |
| | | Pr(>\|t\|) | 0.0002 | | 0.0646 | 0.0646 | |
| | Car/bus (i = 2). | Coeff. | 0.5587 | | 0.0018 | −0.0018 | 0.0099 |
| | | SE | 0.0858 | | 0.0009 | 0.0009 | 0.5914 |
| | | Pr(>\|t\|) | 0 | | 0.0648 | 0.0648 | |
| Denver/Los Angeles (6) | Air (i = 1). | Coeff. | 0.5755 | | −0.0012 | 0.0012 | 0.007 |
| 2003 to 2019 | | SE | 0.0857 | | 0.0006 | 0.0006 | 0.416 |
| | | Pr(>\|t\|) | 0 | | 0.0481 | 0.0481 | |
| | Car/bus (i = 2). | Coeff. | 0.4246 | | 0.0012 | −0.0012 | 0.0098 |
| | | SE | 0.0858 | | 0.0006 | 0.0006 | 0.584 |
| | | Pr(>\|t\|) | 0.0002 | | 0.0481 | 0.0481 | |
| Boston/Chicago | Air (i = 1). | Coeff. | 0.4051 | | −0.0008 | 0.0008 | 0.007 |
| 2004/2019 | | SE | 0.0786 | | 0.0004 | 0.0004 | 0.4219 |
| | | Pr(>\|t\|) | 0.0002 | | 0.0839 | 0.0839 | |
| | Car/bus (i = 2). | Coeff. | 0.5955 | | 0.0008 | −0.0008 | 0.0096 |
| | | SE | 0.0788 | | 0.0004 | 0.0004 | 0.5781 |
| | | Pr(>\|t\|) | 0 | | 0.0843 | 0.0843 | |
| New York/Orlando | Air (i = 1). | Coeff. | 0.4159 | | −0.0014 | 0.0014 | 0.0068 |
| 2003 to 2017 | | SE | 0.097 | | 0.0004 | 0.0004 | 0.4086 |
| | | Pr(>\|t\|) | 0.0011 | | 0.0045 | 0.0045 | |
| | Car/bus (i = 2). | Coeff. | 0.584 | | 0.0014 | −0.0014 | 0.0099 |
| | | SE | 0.097 | | 0.0004 | 0.0004 | 0.5914 |
| | | Pr(>\|t\|) | 0.0001 | | 0.0045 | 0.0045 | |
| Chicago/Orlando | Air (i = 1). | Coeff. | 0.5035 | | −0.0012 | 0.0012 | 0.0068 |
| 2003 to 2017 | | SE | 0.0638 | | 0.0005 | 0.0005 | 0.4086 |
| | | Pr(>\|t\|) | 0 | | 0.0238 | 0.0238 | |
| | Car/bus (i = 2). | Coeff. | 0.4964 | | 0.0012 | −0.0012 | 0.0099 |
| | | SE | 0.0638 | | 0.0005 | 0.0005 | 0.5914 |
| | | Pr(>\|t\|) | 0 | | 0.0237 | 0.0237 | |
| Miami/Washington | Train (i = 1) | Coeff. | 0.0045 | −0.0002 | 0.0002 | 0 | 0.0004 |
| 2007 to 2019 | | SE | 0.0265 | 0.0002 | 0.0002 | 0.0001 | 0.0221 |
| | | Pr(>\|t\|) | 0.867 | 0.4536 | 0.5206 | 0.9988 | |
| | Air (i = 2). | Coeff. | 0.3341 | 0.0002 | −0.001 | 0.0008 | 0.0071 |
| | | SE | 0.1063 | 0.0002 | 0.0006 | 0.0005 | 0.4252 |
| | | Pr(>\|t\|) | 0.0054 | 0.5206 | 0.1039 | 0.1437 | |
| | Car/bus (i = 3). | Coeff. | 0.6614 | 0 | 0.0008 | −0.0008 | 0.0093 |
| | | SE | 0.1073 | 0.0001 | 0.0005 | 0.0005 | 0.5527 |
| | | Pr(>\|t\|) | 0 | 0.9424 | 0.1437 | 0.1383 | |
| New York/Miami | Air (i = 1). | Coeff. | 0.3166 | | −0.0012 | 0.0012 | 0.007 |
| 2003 to 2019 | | SE | 0.1414 | | 0.0005 | 0.0005 | 0.416 |
| | | Pr(>\|t\|) | 0.0419 | | 0.0281 | 0.0281 | |
| | Car/bus (i = 2). | Coeff. | 0.684 | | 0.0012 | −0.0012 | 0.0098 |
| | | SE | 0.1416 | | 0.0005 | 0.0005 | 0.584 |
| | | Pr(>\|t\|) | 0.0003 | | 0.0281 | 0.0281 | |

**Table A5.** *Cont.*

| Routes | | | Car Fares: Gasoline Cost. Bus Fares: Diesel Cost. | | | | Average (t, t−1) Unconditional/ Conditional Budget Shares |
|---|---|---|---|---|---|---|---|
| | | | Marginal Budget Share | Slutsky Coefficients | Slutsky Coefficients | Slutsky Coefficients | |
| Route. Period. | Transport Mode | Values | θi | πi1 Train | πi2 Air | πi3 Car/Bus | ϖit |
| Chicago/St. Louis | Train (i = 1) | Coeff. | 0.0038 | −0.0004 | 0.0002 | 0.0002 | 0.0004 |
| 2007 to 2019 | | SE | 0.0548 | 0.0005 | 0.0005 | 0.0005 | 0.0219 |
| | | Pr(>|t|) | 0.9457 | 0.4748 | 0.6912 | 0.739 | |
| | Air (i = 2). | Coeff. | 0.2697 | 0.0002 | −0.0003 | 0.0001 | 0.0071 |
| | | SE | 0.166 | 0.0005 | 0.0013 | 0.0013 | 0.4225 |
| | | Pr(>|t|) | 0.1208 | 0.6912 | 0.8078 | 0.9249 | |
| | Car/bus (i = 3). | Coeff. | 0.7266 | 0.0002 | 0.0001 | −0.0006 | 0.0093 |
| | | SE | 0.1626 | 0.0005 | 0.0013 | 0.0012 | 0.5556 |
| | | Pr(>|t|) | 0.0002 | 0.739 | 0.9249 | 0.6357 | |
| Buffalo/New York City (7) | Train (i = 1) | Coeff. | 0.0074 | −0.0002 | 0.0002 | 0 | 0.0004 |
| 2006 to 2019 | | SE | 0.0091 | 0.0001 | 0.0001 | 0.0001 | 0.0219 |
| | | Pr(>|t|) | 0.4276 | 0.0935 | 0.1836 | 0.7555 | |
| | Air (i = 2). | Coeff. | 0.939 | 0.0002 | −0.0002 | 0 | 0.0071 |
| | | SE | 0.0363 | 0.0001 | 0.0005 | 0.0006 | 0.4225 |
| | | Pr(>|t|) | 0 | 0.1836 | 0.7931 | 0.9671 | |
| | Car/bus (i = 3). | Coeff. | 0.0536 | 0 | 0 | 0 | 0.0093 |
| | | SE | 0.0372 | 0.0001 | 0.0006 | 0.0006 | 0.5556 |
| | | Pr(>|t|) | 0.1644 | 0.9262 | 0.9671 | 0.9487 | |
| Portland/Seattle | Train (i = 1) | Coeff. | −0.0015 | 0 | −0.0002 | 0.0002 | 0.0004 |
| 2003 to 2019 | | SE | 0.0125 | 0.0001 | 0.0001 | 0.0001 | 0.0219 |
| Note (A) | | Pr(>|t|) | 0.9048 | 0.7046 | 0.1259 | 0.1833 | |
| | Air (i = 2). | Coeff. | 0.6754 | −0.0002 | −0.0012 | 0.0014 | 0.007 |
| | | SE | 0.0804 | 0.0001 | 0.0007 | 0.0007 | 0.4069 |
| | | Pr(>|t|) | 0 | 0.1259 | 0.115 | 0.0609 | |
| | Car/bus (i = 3). | Coeff. | 0.3261 | 0.0002 | 0.0014 | −0.0016 | 0.0098 |
| | | SE | 0.0802 | 0.0001 | 0.0007 | 0.0007 | 0.5713 |
| | | Pr(>|t|) | 0.0004 | 0.2138 | 0.0609 | 0.0367 | 0.0216 |
| Chicago/Detroit (8) | Train (i = 1) | Coeff. | 0.0104 | −0.0002 | 0.0001 | 0.0002 | 0.0004 |
| 2003 to 2018 | | SE | 0.0172 | 0.0002 | 0.0002 | 0.0002 | |
| | | Pr(>|t|) | 0.5504 | 0.2353 | 0.6085 | 0.3286 | |
| | Air (i = 2). | Coeff. | 0.5554 | 0.0001 | −0.0018 | 0.0017 | 0.0069 |
| | | SE | 0.0917 | 0.0002 | 0.0007 | 0.0007 | 0.4049 |
| | | Pr(>|t|) | 0 | 0.6085 | 0.0179 | 0.0229 | |
| | Car/bus (i = 3). | Coeff. | 0.4342 | 0.0001 | 0.0017 | −0.0018 | 0.0098 |
| | | SE | 0.0927 | 0.0002 | 0.0007 | 0.0007 | 0.5734 |
| | | Pr(>|t|) | 0.0001 | 0.4231 | 0.0229 | 0.0147 | |

(6) Denver–Los Angeles. The regressions on air and on car/bus do not meet the homoscedasticity hypothesis: studentized Breusch–Pagan test; data: air, *p* value = 0.0652; car/bus *p* value = 0.065. (7) Train data is the sum of three services: Lake, Maple, and Empire. Air data sum up three routes: Buffalo to NY City, NY City to Rochester, and NY City to Syracuse. (8) Statistical tests say that multi-equation restrictions are not relevant. Introducing them would make the fitting results significantly worse. Likelihood ratio test: Pr (>Chisq) 0.06557. (A) Increased Seattle–Portland train frequency in 2006 to 4 daily and introduced service to Vancouver (BC) in August 2009 [114] (eliminating the need for commuters to alight and connect in Seattle). In December 2017, service frequency was to increase from 4 to 6 per day, but this did not ultimately occur due to a derailment, which also meant service restrictions until spring of the following year and a possible short-term effect on demand [115].

## Appendix C. A Brief Comparative Review in the Scientific Literature

**Table A6.** Elasticities at the national level, unless indicated otherwise: road transport.

| Road | Fuel Price Elasticity Values | Income Elasticity |
|---|---|---|
| Oum et al. (1990) [28] | Car: −0.1 to −1.10. Bus: −0.1 to −1.30. | N.E. |
| Goodwin (1992) [29] | Traffic: −0.16 (short-term), −0.33 (long-term). Bus: −0.41 (−0.28, short-term) | N.E. |
| Johansson and Schipper (1997) [116] | Car: −0.05 to −0.55 (long-term). | Car: 0.65 to 1.25 (long-term). |
| Paulley et al. (2006) [117] | Bus: −0.36 (UK). | Bus: 0 (short−term, UK). −0.15 to −0.63 (long-term). |
| Goodwin et al. (2004) [118] | Personal motor−vehicle: −0.1 (short-term), −0.3 (long-term). | Personal motor−vehicle: 0.2 in the short-term and 0.5 in the long-term (volume of traffic) |
| Holmgren (2007) [30] | Public transport: 0.009 to −1.32 (mean value −0.38). | N.E. |
| Hymel et al. (2010) [119] | Personal motor-vehicle: −0.026 (short-term), −0.135 (long-term) (2004). | 0.5 |
| Escañuela (2019) [33] | −0.45, −0.29 (Northeast Corridor, NEC, route level) | 0.65, 0.42 (NEC, route level) |

**Table A7.** Elasticities at the national level, unless indicated otherwise: air transport.

| Air | Price Elasticity Values | Income Elasticity |
|---|---|---|
| Oum et al. (1990). [28] | From −0.7 to −2.1. | N.E. |
| Brons et al. (2002) [120] | −1.146 (although there are estimates from 0.21 to −3.20) | N.E. |
| Kincaid and Trethaway (2007) [121] | Route level, short-haul: −1.54, long-haul: −1.40. National level: (−0.88, −0.80) | N.E. |
| Smyth and Pearce (2008) [122] | Route Level: −1.4 (−1.54 short-haul routes). National Level: −0.8 (−0.88 short-haul routes) | 1.8 to 2.2. (Depending on route length) |
| Chi et al. (2012) [123] | −1.2 a −1.5 (2000), −2.5 to −3.3 (2005) (p. 89) | N.E. |
| Clewlow et al. (2014) [124] | By route (Europe), elasticity with respect to jet fuel price: −1.863 to −2.304. | N.E. |
| Gundelfinger (2018) [31] | −0.62 | 0.81 |
| Escañuela (2019) [33] | −0.73, −0.84 (NEC, route level) | 1.55, 1.88 (NEC, route level) |

**Table A8.** Elasticities. Rail passenger transport.

| Rail | Price Elasticity Values | Income Elasticity |
|---|---|---|
| Jones and Nichols (1983) [125] | (UK) −0.64 | N.E. |
| Doi and Allen (1986) [126] | (US) −0.245 | N.E. |
| Oum et al. (1990) [28] | Rail intercity −0.30 to −1.18 | N.E. |
| Goodwin (1992) [29] | −0.79 | N.E. |
| Douglas and Karpouzis (2009) [127] | (Sydney metropolitan rail, 1969–2008, annual data, parameters with the expected sign, but none of them significant at the 95% confidence level except the constant) −0.283 | (Real GDP pc) 0.74 |
| Hortelano et al. (2016) [128] | (Short-term, Spain, high-speed rail) −0.6 | N.E. |

**Table A8.** *Cont.*

| Rail | Price Elasticity Values | Income Elasticity |
|---|---|---|
| Brumerčíková et al. [129] | (Slovak Rep., cross price elasticity, oil prices, depending ob the year) positive and negative elasticities | (Slovak Republic, depending on the year) positive and negative elasticities |
| Escañuela (2019) [33] | −0.44, −0.47 (NEC, route level) | 0.13, 0.20 (NEC, route level) |
| Zeng et al. (2021) [25] | (China) −1.049 to −1.090; (China, cross-price elasticities of demand, train–air, train–car) approx. 0.001 | N.E. |
| Wijeweera and Charles (2023) [130] | (Australia, Melbourne) −0.07 | N.E. |

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
