# Peer review of "Elasticities of Passenger Transport Demand on US Intercity Routes: Impact on Public Policies for Sustainability"

_sustainability, doi:10.3390/su151814036_

Round 1
Reviewer 1 Report
The topic is very challenging and the arguments offered by authors are convincing in debating it.
The introduction describes the theoretical background, by briefly presenting the context of this paper. The cited references are relevant to the research. However, the authors could have expanded more the literature review part.
Even though the authors mentioned the three objectives of the paper (“i) to contribute to estimating the elas-101 ticities of demand in passenger transport for domestic routes in the United States, where 102 there is competition between air, road, and rail transport; ii) to contribute to the compari-103 son of those estimates on the different routes analysed from 2003 to 2019 in order to check 104 whether there are relevant variations between them (which requires comparison on a 105 common basis); and iii) to draw some conclusions based on the relationship between the 106 estimated demand elasticities and the effectiveness of different public policies to be im-107 plemented to improve the sustainability of passenger transport in the United States on 108 domestic routes”), they did not clearly stated the purpose of their research (which should be mentioned even in the abstract).
The methods that were used are adequately stated and the results are presented in clear manner, the authors offering coherent and compelling arguments.
The authors could develop more the last part, entitled “discussion”, also by adding some conclusions.
The paper needs English proof-reading.
Author Response
For the attention of Reviewer 1
Dear reviewer,
Please find attached the report with the modifications introduced in the analysis and included in the revised version of the article in order-respond-your suggestions. Thank you for your suggestions and questions.
We have carefully analysed and responded-each of them. This has resulted in important modifications-the original version, which we believe has been improved. We hope we have satisfied all your requests and remain at your disposal-answer any questions you may have.
Sincerely,
The authors
Below is a list of the questions and suggestions raised and their answers.
When some questions are related, we include them in succession and try-give an overall answer at once (to facilitate reading and avoid repeated explanations).
Issue 1. Is the content succinctly described and contextualized with respect-previous and present theoretical background and empirical research (if applicable) on the topic? Literature review / purpose (appart 3 purposes)
The references in the manuscript have been expanded and revised in all sections. In addition, content has been added in lines 47-51 and 66-72-contextualise the aims of the manuscript. Content has also been added in lines 94-110, 119-127 and 136-140-provide more theoretical context and-substantiate the choice of method made. Finally, the choice of method has been further contextualised (lines 184-192).
Issue 2. Are the arguments and discussion of findings coherent, balanced and compelling?
Some considerations have been added-the Results,-clarify them (lines 457-459, 462-464) or-explain the limitations of the results obtained (lines 457-459, 524-528, Appendix B - Table 2). In addition, modifications have been introduced in the Discussion section-clarify the results, their justification and implications (lines 541-569, 574-578, 592-595, 620-621, 631-632). In general, the objectives set are achieved (and some of the implications of the estimated elasticities are developed), except for the statistical significance of the coefficients referring-the rail mode of transport. Possible reasons for this limitation of the study are discussed.
Issue 3. Are the conclusions thoroughly supported by the results presented in the article or referenced in secondary literature?
As indicated above, the references in the manuscript have been expanded and revised in all sections. Also, some considerations have been added-the Results,-clarify them (lines 457-459, 462-464) or-explain the limitations of the results obtained (lines 457-459, 524-528, Appendix B - Table 2).
Issue 4. In the introduction: However, the authors could have expanded more the literature review.
The references in the manuscript have been expanded and revised. Furthermore, content has been added in lines 47-51 and 66-72-contextualise the objectives of the manuscript. Also, content has been added in lines 94-110, 119-127 and 136-140-provide more theoretical context and further rationale for the choice of method made.
Issue 5. They did not clearly stated the purpose of their research (which should be mentioned even in the abstract).
The abstract (lines 13-17,21-29) has been modified-more clearly include the objectives of the manuscript. Moreover, the wording of the objectives has been modified-specify one general objective and three instrumental objectives (lines 141-156).
Issue 6. Develop more the last part, entitled “discussion”, also by adding some conclusions.
In the Discussion section, modifications have been introduced-clarify the results, their justification and their implications (lines 541-569, 574-578, 592-595, 620-621, 631-632). In general, the objectives set are achieved (and some of the implications of the estimated elasticities are developed), except for the statistical significance of the coefficients referring-the rail mode of transport. Possible reasons for this limitation of the study are discussed.
English.
The manuscript has been submitted-the English-language review service.

Reviewer 2 Report
Estimating the demand elasticity of goods is an essential task in economics. The paper estimates the Rotterdam Demand Model using the Seemingly Unrelated Regression Method for transportation sector. Authors explained in detail about the chosen method. The method is appropriate to the topic. The data and methodology are explained in detail. Elasticity coefficients are estimated for air and road for different routes. Exploring heterogenous results is interesting. The paper has a contribution to the existing literature in terms of estimating elasticities for transportation sector.
Authors may go over the language of the text.
Author Response
For the attention of Reviewer 2
Dear Reviewer,
Some changes have been made to improve the manuscript. They have been highlighted.
Thank you very much.
Sincerely,
The authors
English
The manuscript has been submitted to the English-language review service.
Reviewer 3 Report
The study investigates Elasticities of passenger transport demand on US intercity routes: impact on public policies for sustainability. By understanding the elasticity of demand, the study estimates and predicts the intensity of transportation demand and the effects of various policies aimed at reducing greenhouse gas emissions. The paper demonstrates clear thinking, a well-structured approach, and holds practical value. Below are some strengths and weaknesses of the paper:
1.The paper presents a solid theoretical foundation and methodology, and supports the author's hypotheses through clear empirical results.
2.The methods and experimental design in the paper display a considerable level of credibility and replicability, providing a foundation for further research in related fields.
3.The research methods in the paper have certain limitations, such as sample size or difficulties in data collection, which may affect the generalizability of the results.
4.Overall, this paper addresses a valuable research question and contributes to the field's development through innovative methods and in-depth empirical analysis.
Author Response
For the attention of Reviewer 3
Dear reviewer,
Please find attached the report with the modifications introduced in the analysis and included in the revised version of the article in order-respond-your suggestions. Thank you for your suggestions and questions.
We have carefully analysed and responded-each of them. This has resulted in important modifications-the original version, which we believe has been improved. We hope we have satisfied all your requests and remain at your disposal-answer any questions you may have.
Sincerely,
The authors
Issue 1. Is the content succinctly described and contextualized with respect-previous and present theoretical background and empirical research (if applicable) on the topic?
The references in the manuscript have been expanded and revised in all sections. In addition, content has been added in lines 47-51 and 66-72-contextualise the aims of the manuscript. Content has also been added in lines 94-110, 119-127 and 136-140-provide more theoretical context and-substantiate the choice of method made. Finally, the choice of method has been further contextualised (lines 184-192).
Issue 2. Are all the cited references relevant-the research?
As mentioned above, the references in the manuscript have been expanded and revised in all sections.
Issue 3. Are the research design, questions, hypotheses and methods clearly stated?
The abstract (lines 13-17,21-29) has been modified-more clearly include the aims and methods of the manuscript. In addition, content has been added in lines 47-51 and 66-72-contextualise the aims of the manuscript. Content has been added in lines 94-110, 119-127 and 136-140-provide more theoretical context and further justification for the choice of method made. Finally, the wording of the objectives has been modified-specify one general objective and three instrumental objectives (lines 141-156). The choice of method has also been further contextualised (lines 184-192).
Issue 4. Are the arguments and discussion of findings coherent, balanced and compelling?
Some considerations have been added-the Results-clarify them (lines 457-459, 462-464) or-explain the limitations of the results obtained (lines 457-459, 524-528, Appendix B - Table 2). Furthermore, modifications have been introduced in the Discussion section-clarify the results, their justification and implications (lines 541-569, 574-578, 592-595, 620-621, 631-632). In general, the objectives set are achieved (and some of the implications of the estimated elasticities are developed), except for the statistical significance of the coefficients referring-the rail mode of transport. Possible reasons for this limitation of the study are discussed.
Issue 5. For empirical research, are the results clearly presented?
Some considerations have been added-the Results,-clarify them (lines 457-459, 462-464) or-explain the limitations of the results obtained (lines 457-459, 524-528, Annex B - Table 2).
Issue 6. Is the article adequately referenced?
The references in the manuscript have been expanded and revised in all sections.
Issue 7. Are the conclusions thoroughly supported by the results presented in the article or referenced in secondary literature?
As indicated above, the references in the manuscript have been expanded and revised in all sections. Some considerations have been added-the Results,-clarify them (lines 457-459, 462-464) or-explain the limitations of the results obtained (lines 457-459, 524-528, Appendix B - Table 2). Likewise, in the Discussion section, modifications have been introduced-clarify the results, their justification and their implications (lines 541-569, 574-578, 592-595, 620-621, 631-632). In general, the objectives set are achieved (and some of the implications of the estimated elasticities are developed), except for the statistical significance of the coefficients referring-the rail mode of transport. Finally, the possible reasons for this limitation of the study are discussed.
Issue 8. The research methods in the paper have certain limitations, such as sample size or difficulties in data collection, which may affect the generalizability of the results.
Some considerations have been added-the Results-clarify them (lines 457-459, 462-464) or-explain the limitations of the results obtained (lines 457-459, 524-528, Appendix B - Table 2). In addition, modifications have been introduced in the Discussion section-clarify the results, their justification and implications (lines 541-569, 574-578, 592-595, 620-621, 631-632). In general, the objectives set are achieved (and some of the implications of the estimated elasticities are developed), except for the statistical significance of the coefficients referring-the rail mode of transport. Possible reasons for this limitation of the study are discussed.
English.
The manuscript has been submitted-the English-language review service.

Round 2
Reviewer 3 Report
Thanks for the efforts of responding my comments. You have addressed my concerns to a satisfactory level.